# Estimating Vertical Ground Reaction Force during Walking Using a Single Inertial Sensor

**DOI:** 10.3390/s20154345

**Published:** 2020-08-04

**Authors:** Xianta Jiang, Christopher Napier, Brett Hannigan, Janice J. Eng, Carlo Menon

**Affiliations:** 1Menrva Research Group, Schools of Mechatronic Systems & Engineering Science, Simon Fraser University, Metro Vancouver, BC V3T 0A3, Canada; xiantaj@mun.ca (X.J.); cnapier@sfu.ca (C.N.); bchannig@sfu.ca (B.H.); 2Department of Computer Science, Memorial University of Newfoundland, St. John’s, NL A1B 3X5, Canada; 3Department of Physical Therapy, University of British Columbia, Vancouver, BC V6T 1Z3, Canada; janice.eng@ubc.ca; 4Rehabilitation Research Program, GF Strong Rehab Centre, Vancouver Coastal Health Research Institute, Vancouver, BC V5Z 2G9, Canada

**Keywords:** IMU, ground reaction force, gait analysis, walking

## Abstract

The vertical ground reaction force (vGRF) and its passive and active peaks are important gait parameters and of great relevance for musculoskeletal injury analysis and prevention, the detection of gait abnormities, and the evaluation of lower-extremity prostheses. Most currently available methods to estimate the vGRF require a force plate. However, in real-world scenarios, gait monitoring would not be limited to a laboratory setting. This paper reports a novel solution using machine learning algorithms to estimate the vGRF and the timing and magnitude of its peaks from data collected by a single inertial measurement unit (IMU) on one of the lower limb locations. Nine volunteers participated in this study, walking on a force plate-instrumented treadmill at various speeds. Four IMUs were worn on the foot, shank, distal thigh, and proximal thigh, respectively. A random forest model was employed to estimate the vGRF from data collected by each of the IMUs. We evaluated the performance of the models against the gold standard measurement of the vGRF generated by the treadmill. The developed model achieved a high accuracy with a correlation coefficient, root mean square error, and normalized root mean square error of 1.00, 0.02 body weight (BW), and 1.7% in intra-participant testing, and 0.97, 0.10 BW, and 7.15% in inter-participant testing, respectively, for the shank location. The difference between the reference and estimated passive force peak values was 0.02 BW and 0.14 BW with a delay of −0.14% and 0.57% of stance duration for the intra- and inter-participant testing, respectively; the difference between the reference and estimated active force peak values was 0.02 BW and 0.08 BW with a delay of 0.45% and 1.66% of stance duration for the intra- and inter-participant evaluation, respectively. We concluded that vertical ground reaction force can be estimated using only a single IMU via machine learning algorithms. This research sheds light on the development of a portable wearable gait monitoring system reporting the real-time vGRF in real-life scenarios.

## 1. Introduction

The vertical ground reaction force (vGRF) is the largest component of the ground reaction force during walking, resulting in forces greater than one body weight (BW) per step. The curve of the vGRF consists of two peaks: the passive (weight acceptance) peak and the active (push off) peak. The passive peak (PP) is the result of the foot’s collision with the ground, whereas the active peak (AP) results from the active force applied by the foot into the ground as it pushes off. The magnitude and timing of these peaks influence the loads experienced at the joints and muscles of the lower limb, and may result in the development or exacerbation of musculoskeletal overuse injuries and conditions such as osteoarthritis [1,2]. Typically, the measurement of the vGRF requires a force plate, either embedded in an instrumented treadmill or in an over ground walkway. The downsides to this method of measurement include the cost of equipment, spatial requirements, and the limitations of testing in a lab environment as opposed to a natural (“in field”) environment.

Advances in wearable technology have made it possible to quantify kinetic parameters in clinical or “in field” environments, but the estimation of the vGRF has yet to approach gold standard laboratory measurements [3]. Several wearable devices have been developed and offer promise for estimating the vGRF [3]. Load cells [4,5], pressure sensor array [6], and pressure insoles [7,8] have all been used. Load cells are not a practical or cost-effective solution, while pressure sensors and pressure insoles have disadvantages, such as low durability and the need for calibration. Previous studies have shown a correlation between peak vertical/axial accelerations measured with body-worn accelerometers and the peak vGRF in running [9,10,11]. Indirect estimation of the peak vGRF also allows one to estimate individual joint loading characteristics via inverse dynamics. Karatsidis et al. [12] recently reported the use of a set of inertial measurement units (IMUs) to estimate ground reaction forces and moments. This approach was successful, but still requires musculoskeletal modeling and calibration before use. It also requires a large set of IMUs, which may be expensive and impractical in a non-laboratory setting. Shahabpoor et al. [13] used a single IMU fixated at the 7th cervical vertebra to estimate the vGRF using a “Scaled Acceleration” (SA) method. This method was able to estimate the vGRF to just over 5% normalized mean squared error (NRMSE) and was able to reliably predict the timing of gait events and vGRF amplitudes.

With the wider adoption of machine learning approaches to the monitoring of human movement using wearable sensors, it is possible to estimate ground reaction forces from IMUs using several different machine learning algorithms. Neural network modeling has already demonstrated its usefulness in gait analysis [14,15] and has also been used to estimate the vGRF based on measurements using pressure insoles [16,17]. Ngoh et al. [18] successfully investigated the use of neural network modeling to estimate the vGRF during running using a shoe-mounted IMU. Random forest learning model is another alternative machine learning approach. The random forest learning algorithm comprises an ensemble of prediction trees in which each tree is trained based on a random selection of part of the training dataset and the best result is selected from the results produced by the trees. Random forest model is robust to outliers and nonlinear and unbalanced data, as well as to low bias and moderate variance [19,20]. The authors have applied a random forest model to accurately estimate ankle joint power using two IMUs on the foot and shank, respectively [21].

Advances in sensors and in machine learning methods have made it possible to approach gold standard (force plate) estimation of the timing and magnitude of the vGRF during walking [22,23]. However, the optimal location of a single IMU for estimation of the vGRF is unknown, as IMU orientation and placement would affect the estimation of the vGRF [24,25]. Therefore, we aimed to determine (i) whether a single inertial sensor per lower limb was able to estimate the magnitude and timing of the vGRF peaks during walking using a random forest method and (ii) the optimal placement of a single IMU.

## 2. Materials and Methods

### 2.1. Experimental Setup

Each participant was outfitted with four wireless IMUs (MTw Awinda, Xsens, Enschede, The Netherlands) mounted on the foot, distal shank, distal thigh, and proximal thigh to record the acceleration and angular velocity of each segment during walking trials (Figure 1). The signals from each of the four IMUs were used to train and test a machine learning model for vGRF estimation. To capture the reference vGRF from the participants while walking, a Bertec split-belt force plate-instrumented treadmill (Bertec Corporation, Columbus, OH, USA) was used. The data from the IMUs and the reference vGRF from the treadmill were synchronized by two synchronization impulse signals generated at the beginning and end of the trial, respectively. The data from both IMUs and treadmill were sampled at 100 Hz.

### 2.2. Protocol and Procedure

Nine healthy, male adults (age 27 ± 8; height 176 ± 7 cm; weight 72 ± 9 kg) were recruited from the local student population for this study. The participants wore their own regular walking shoes and walked on the treadmill at five different speeds: 0.4 m/s, 0.7 m/s, 1.0 m/s, 1.3 m/s, and 1.6 m/s. These speeds covered the normal range of walking speeds for an adult human [26]. Each trial lasted for 1 min, and progressed from low to high speeds. We did not randomize the speeds order so that the participant could anticipate the next speed to walk. Participants were given time to warm up and become comfortable on the treadmill before any data collection commenced. Data collection commenced once the treadmill reached the target speed for 1 min after the initial acceleration period. Participants were given the opportunity to rest in between consecutive trials if they wished.

The study protocol was approved by the Office of Research Ethics at Simon Fraser University, and all participants provided informed consent.

### 2.3. Data Analysis

#### 2.3.1. Data Processing

First, the IMU data underwent a median filter of a five-sample window, and then went through a feature extraction procedure using a 110-ms window (corresponding to 11 samples at a 100 Hz sampling rate). Ten temporal domain features were employed, which included root mean square, sum of absolute value, mean absolute deviation, variance, wave length, slope sign changes, and simple square integral, mean wavelet with db7, difference absolute standard deviation value, average amplitude change, log detector, and the coefficients of linear fit and parabolic fit [27,28]. We chose these features due to their simplicity and undemanding calculation power. They have also previously been proven effective for gait phase partitioning and ankle joint power estimation [21,29]. The extracted features were further normalized to a uniformed range between 0 and 1.

Second, the derived IMU feature data were used to train and test a random forest regression model. The random forest model combined a set of decision trees, each trained on a slightly different set of data randomly selected from the training dataset. The output of the model was majority voted from the results produced by these trees [19,20]. The random forest regressor algorithm is available in the Statistics and Machine Learning Toolbox^TM^ in MATLAB R2019a (The MathWorks, Inc., Natick, MA, USA). The tunable parameters—the numbers of trees in each iteration and the leaves of each tree—were set to fifty and five, respectively.

The derived vGRF was low pass filtered at a 10 Hz cut-off frequency, as the majority of vGRF signal was contained below 10 Hz [30]. The passive peak (PP) was detected by searching local maxima in the first half of each stance cycle, while the active peak (AP) was determined as the local maximum in the second half of each stance cycle. The high frequency component of the vGRF was derived by high pass filtering at a 10 Hz cut-off frequency for illustration purposes.

#### 2.3.2. Evaluation

The performance of the vGRF estimation method was thoroughly assessed in both intra-participant and inter-participant testing settings. Intra-participant testing may also be called same-user testing, where the model is trained and tested using datasets from the same participant. In the present study, the IMU data from each speed were separated into two equal parts in time sequence; the first half of the data of all five speeds were used for model training and the second half of the data for testing. Intra-participant testing usually achieves better accuracy than inter-participant testing but requires training a separate model for each user in real-life applications. Conversely, inter-participant testing trains a model based on the data from a group of participants separate from the data of the participant to be tested. For inter-participant testing in the present study, the IMU data at all five speeds from one of the nine participants were selected as testing data, and IMU data from the remaining eight participants at all five speeds were assigned to the training dataset. This process was repeated until the data from each participant had been used as testing data. The accuracy was then calculated by averaging the accuracies obtained across all participants at all walking speeds. The inter-participant testing evaluates the generalization ability of the model when applied to a new user without further model training.

Multiple performance measures were employed in this study, including the correlation coefficient (R), the root mean squared error (RMSE), and the normalized root mean square error (NRMSE). R in the present study measured the similarity between the estimated vGRF values and their references. The measurement values of R range from −1.0 to 1.0, where R = 1.0 represents a perfectly positive correlation, R = 0.0 expresses no linear relationship between the two variables, and R = −1.0 is a perfectly negative correlation. The RMSE measures the overall distance between the estimated and reference vGRF trajectories, and the NRMSE normalizes the RMSE by dividing the RMSE range of the reference vGRF. R, the RMSE, and the NRMSE were calculated for each participant throughout the whole course, and the mean and standard deviation were calculated across all participants.

The peak values of the vGRF (PP and AP) and their occurrence (timing) during each stance phase were also recorded. As the peak values were determined within each stance phase, we defined the start and end of a stance phase (initial contact and toe-off) by a vGRF threshold of 20 N. The same procedure was repeated for all four IMUs.

We performed a two-way analysis of variance (ANOVA) to examine the effect of the independent variables—sensor location and speed—on the dependent variables of correlation coefficient (R), the RMSE, the NRMSE, and passive/active peak error and delay, in both intra- and inter-participant scenarios. Post hoc pairwise comparisons, Tukey’s HSD (honestly significant difference), were further conducted if there were any significant effects of the independent variables on the dependent variables. The significance level was set to *p* < 0.05.

## 3. Results

All nine participants completed the entire data collection protocol. As each of five speed trials per participant lasted for approximately 1 min and data were collected at 100 Hz from both the IMUs and treadmill, a total of 30,000 IMU data points were collected from each participant, resulting in 270,000 samples collected across 45 recorded trials from all nine participants. We excluded seven trials because the ground reaction force was not recorded correctly (e.g., when the participants sometimes walked on only one of the treadmill belts). A sample of collected IMU signals and the corresponding reference vGRF in the gait cycles is shown in Figure 2. An example of reference and estimated vGRF profiles with the corresponding low and high frequency components for a participant walking at 1.6 m/s is shown in Figure 3.

Figure 3 shows intra- and inter-participant testing examples of the estimated vGRF compared to the reference vGRF and their decomposition into low (≤10 Hz) and high (>10 Hz) frequency components, for a participant walking at a speed of 1.6 m/s, using an IMU on the shank.

The accuracies for the regression models of all four locations (foot, shank, distal thigh, and proximal thigh) in intra- and inter-participant testing are listed in Table 1 and Table 2, respectively.

Table 3 and Table 4 show the intra-participant and inter-participant testing results, respectively, from all speeds averaged across all IMUs.

The two-way ANOVA for the intra-participant testing showed that there was neither a significant effect of sensor location nor speed on the correlation coefficient (R), RMSE, or NRMSE. There were significant effects of both sensor location and speed on the PP error (location, *F* (3150) = 3.04, *p* < 0.05; speed, *F* (4150) = 7.35, *p* < 0.001, respectively). There was no significant interaction effect between speed and location on PP error. The post hoc test on the effect of location showed that there was only a significant difference in PP error between the foot location and the shank (*p* < 0.05), and there was no significant difference between the shank and thighs. The post hoc test on the effect of speed showed that there was only a significant difference in PP error between the highest speed (1.6 m/s) and lower speeds (*p* < 0.05). There was significant effect of speed (*F* (4150) = 8.7, *p* < 0.05) but no significant effect of location on the PP delay. The post hoc test on the effect of speed showed that there was only a significant difference in PP delay between speed 0.4 m/s and speeds 1.0 m/s, 1.3 m/s, and 1.6 m/s (*p* < 0.05).

There was only significant effect of IMU location on AP error (*F* (3150) = 13.71, *p* < 0.001). The post hoc test on the effect of location showed that only the IMU on the foot was significantly different from other locations (*p* < 0.001), but there was no significant difference between the other locations. There was only significant effect of speed on AP delay (*F* (4150) = 4.78, *p* < 0.005). The post hoc test on the effect of speed showed that only the AP delay of speed 1.0 m/s was different from speeds 0.4 m/s, 1.3 m/s, and 1.6 m/s, but there was no significant difference between these other four speeds.

The two-way ANOVA for the inter-participant testing showed that there were significant effects of IMU location on R (*F* (3150) = 4.36, *p* < 0.01), RMSE (*F* (3150) = 5.38, *p* < 0.005) and NRMSE (*F* (3150) = 5.2, *p* < 0.005), but no significant effect of speed. The post hoc test (Tukey’s HSD) on the effect of location on R, RMSE, and NRMSE showed that there were only significant differences (*p* < 0.05) between the IMUs on the shank and distal thigh.

There were both significant effects of sensor location and speed on the PP error (location, *F* (3150) = 3.63, *p* < 0.05; speed, *F* (4150) = 3.87, *p* < 0.001, respectively). There was no significant interaction effect of speed and location on PP error. The post hoc test on the effect of location showed that there was only a significant difference in PP error between the foot location and the shank (*p* < 0.05), and there was no significant difference between the shank and thighs. The post hoc test on the effect of speed showed that there was only a significant difference in PP error between the highest speed (1.6 m/s) and two of the lower speeds (1.0 m/s, 1.3 m/s, *p* < 0.05). There was significant effect of speed (*F* (4150) = 18.23, *p* < 0.001) but no significant effect of location on the PP delay. The post hoc test on the effect of speed showed that there was a significant difference in PP delay between speeds 0.4 m/s and 0.7 m/s and speeds 1.0 m/s, 1.3 m/s, and 1.6 m/s (*p* < 0.05); there was also a significant difference between speeds 0.4 m/s and 0.7 m/s (*p* < 0.05), but there was no significant difference between speeds 1.0 m/s, 1.3 m/s, and 1.6 m/s.

There was neither a significant effect of speed nor IMU location on the AP error. There was only a significant effect of speed on AP delay (*F* (4150) = 14.06, *p* < 0.001). The post hoc test on the effect of speed showed that the AP peak appeared significantly earlier in the lowest speed (0.4 m/s) compared to all other speeds (*p* < 0.05), but there was no significant difference between the other four speeds (0.7–1.6 m/s).

Figure 4 shows the mean peak amplitude of the reference and estimated the vGRF (passive and active force peaks) across all nine participants for each speed, in intra-participant testing (Figure 4A) and inter-participant testing (Figure 4B), respectively.

## 4. Discussion

In this study, we investigated the performance of a single IMU to estimate the vertical ground reaction force (vGRF) and the timing and magnitude of force peaks during walking. A random forest algorithm was used to estimate vGRFs in intra- and inter-participant scenarios. In both scenarios, the IMU on the shank outperformed the other locations. High accuracies were achieved at the shank location, with correlation coefficients (R) of 1.00 and 0.97 between the estimated vGRF and references in intra- and inter-participant scenarios, respectively. Low root mean square errors were also achieved with 0.02 BW and 0.10 BW in these two scenarios, respectively. The high correlation and low error demonstrate the effectiveness of a single IMU located on the shank for sample-by-sample-based precise vGRF estimation during normal walking speeds.

The magnitude and timing of the force peaks are also important parameters in gait analysis. The passive peak (PP) is caused by weight acceptance as the participant transitions from double-limb to single-limb support. Accurately estimating the PP is critical for applications such as reducing musculoskeletal injuries. In this study, low magnitude errors of 0.02 BW and 0.14 BW were achieved for the PP in intra- and inter-participant scenarios, respectively, with a delay of −0.14% and 0.57% of total stance duration. Similar low errors in magnitude and timing were achieved in active peak (AP) estimation. The results demonstrate the effectiveness of a single IMU to accurately estimate the magnitude and timing of vGRF peaks in walking.

The number and location of sensors used for gait analysis are critical when measuring gait parameters, in order to improve the effectiveness and convenience of the methods in clinical or “in field” situations. Using a single IMU for measuring gait parameters has significant advantages in terms of cost and practical application. Our findings provide evidence that using a single IMU for vertical ground reaction force (vGRF) estimation is a robust alternative to force plate measurement [13,22,23]. We examined four possible locations for an IMU on the lower limb to determine the best location for this estimation. Interestingly, the ANOVA results showed little difference between locations for estimating the vGRF in intra-participant scenarios; only AP delay was affected by the IMU location. However, placing the IMU on the shank achieved a better RMSE and NRMSE than when fixed to the distal thigh in inter-participant scenarios, and the IMU on the foot achieved better peak errors than the other locations. Overall, the shank location proved to be the most accurate across all speeds and scenarios. This might have been due to lower soft tissue movement artefacts at the shank location compared to the thigh locations, or poor fixation on the shoes. Proximity to the ground (i.e., less opportunity for force attenuation via variation in joint stiffness) may also have improved accuracy.

The choice of modeling algorithm is an important consideration when using a machine learning approach. Ngoh et al. [18] investigated the use of neural network modeling to estimate the vGRF during running using a shoe-mounted IMU. Mean errors in the magnitude were between 0.10 and 0.18 BWs for the passive and active peaks, with timing errors of between 0.34% and 1.3% of stance phase. Studies that have used machine learning approaches to estimate the vGRF in walking have reported errors of between 3.8 and 4.8% [22,23]. These contrast with our results of 1.70% for the shank-mounted IMU in our study, where a random forest ensemble-based machine learning algorithm was employed to estimate the vGRF from a single IMU. More specifically, the random forest algorithm is a bagging technique that partially samples from the input data for a certain range of frames and votes up the best estimation; it is robust to data outliers and variances, which improves the accuracy. Our approach has achieved one of the best performances when compared to other methods in the literature, including Newton’s law of motion, biomechanical models, and other machine learning methods [3].

It is worthwhile noting that the inter-participant evaluation adds to the practicality of the system, since a wearable device based on the proposed method should contain a pre-trained model before delivery to the end user. This is an essential feature for practical scenarios, in which the end users generally do not have access to motion capture or a force plate-instrumented treadmill to obtain the reference data required to train the model. However, even in the intra-participant scenario in this study, we employed a restricted testing method to use the first half of a participant’s data for model training and the second half for model testing, instead of using a randomized cross-validation method. This method proved to be very robust and could be used to calibrate the model to the individual if access to a force plate exists.

## 5. Limitations and Future Work

This study was limited to a lab setting using a force plate-instrumented treadmill for the purpose of model validation. Future work will explore walking outdoors, including different terrain (steps, hills, etc.). Walking at higher speeds—including running speeds—will also be considered in the next stage of this study.

The participants of this study only included young males. A more varied healthy population with a wider range of height, weight, age, and sex will be considered to boost the model prediction accuracy in the future study. A further step to validate the model on clinical populations has also been included in our research plan.

## 6. Conclusions

This paper presents a method to estimate vertical ground reaction force and the magnitude and timing of its peaks using only a single IMU. The advantage of this method is that it can be applied to “in field” and clinical situations. Nine participants were asked to wear four IMUs on their foot, shank, distal thigh, and proximal thigh while walking on a treadmill at different speeds. A random forest regressor was employed to map one of the four IMU signals to vGRF and peak values. This method achieved a very strong correlation coefficient (R = 1.00) averaged over all speeds and an RMSE = 0.02 BW. The timing of the passive peak was correctly identified within −0.1% to 0.6% and the active peak was correctly detected within 0.5% to 1.7% of stance duration in both intra- and inter-participant scenarios. The results of this study suggest the feasibility of using a single IMU to estimate the vertical ground reaction force during walking.

## Figures and Tables

**Figure 1 sensors-20-04345-f001:**
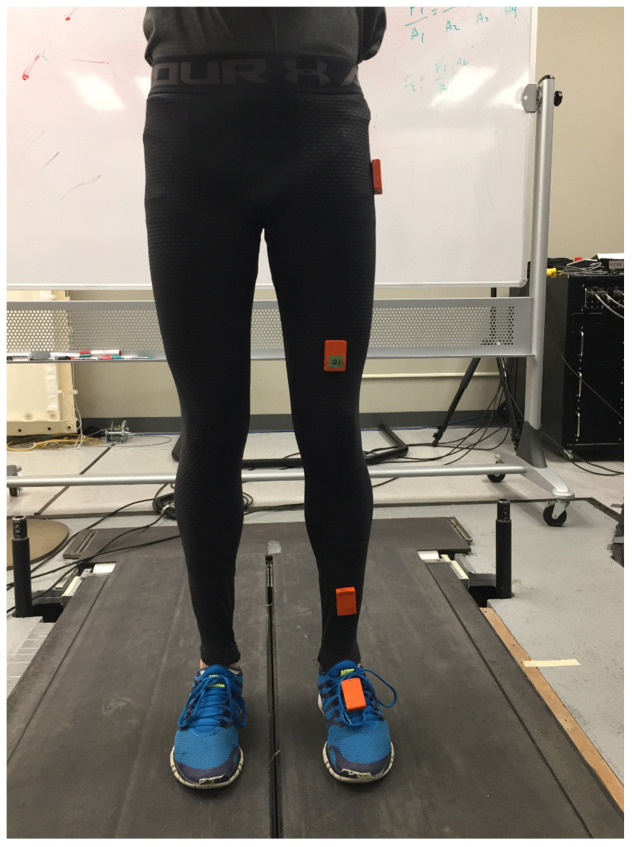
The experimental setup. The participant wears four inertial measurement units (IMUs) on the foot, distal shank, distal thigh, and proximal thigh, walking on a force plate-instrumented treadmill.

**Figure 2 sensors-20-04345-f002:**
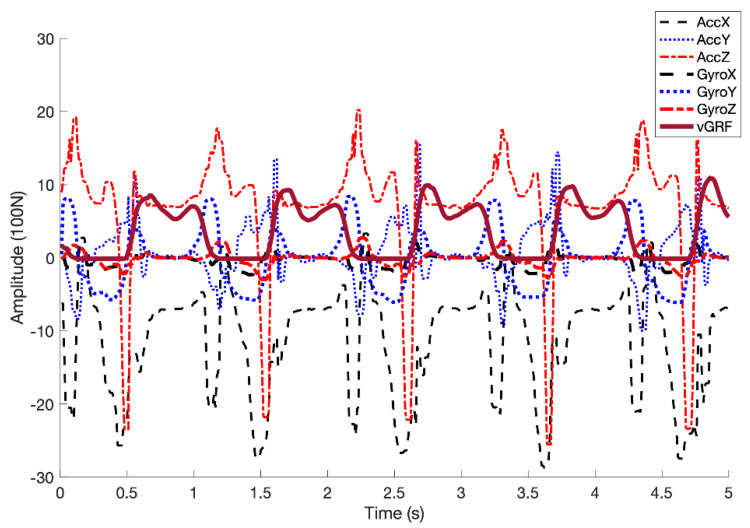
Sample of collected raw IMU signals for a segment including four gait cycles, overlaid with reference vertical ground reaction force (vGRF) labels (bold purple line).

**Figure 3 sensors-20-04345-f003:**
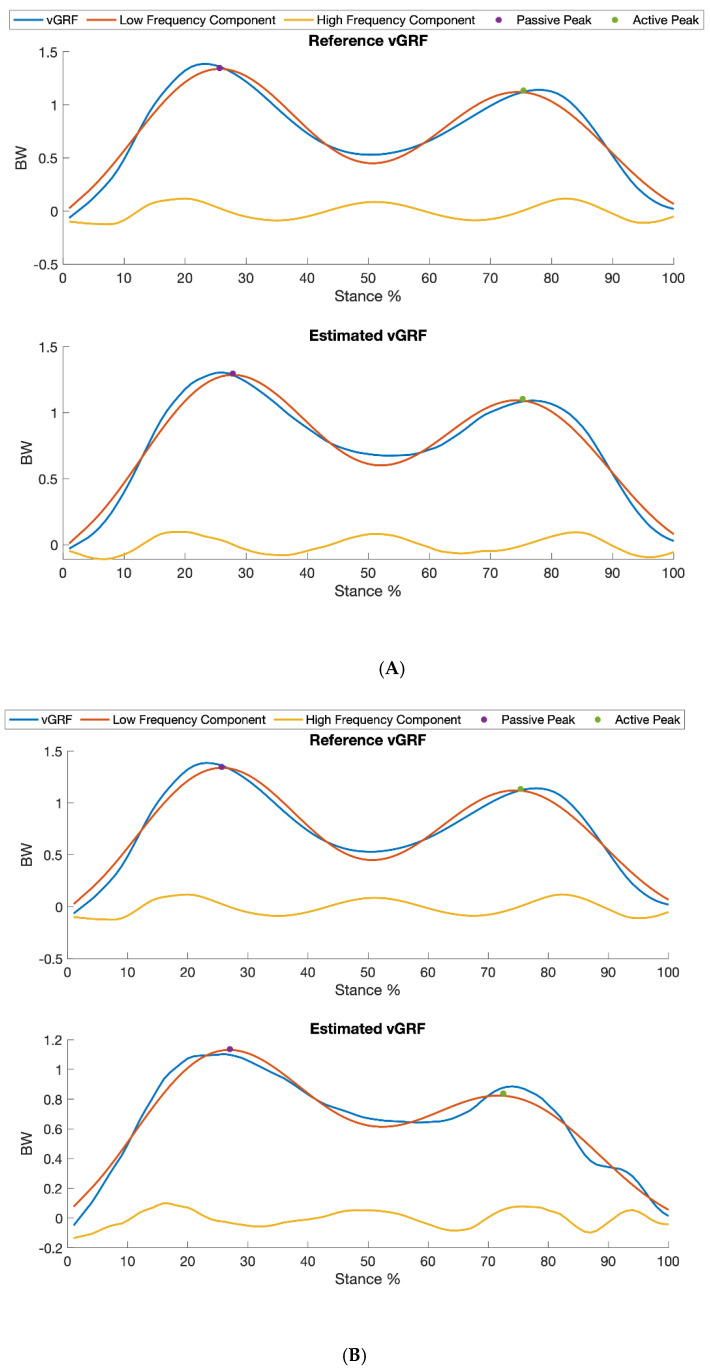
Examples of intra-participant (**A**) and inter-participant (**B**) testing results, showing both the estimated vGRF compared to the reference vGRF and their decomposition into low (≤10 Hz) and high frequency (>10 Hz) force components, for a participant walking at a speed of 1.6 m/s, using an IMU on the shank. In both upper and lower panels, the blue curves represent vGRF profiles, and the red and yellow curves represent low and high frequency forces, respectively. The amplitude is body weight (BW) scaled.

**Figure 4 sensors-20-04345-f004:**
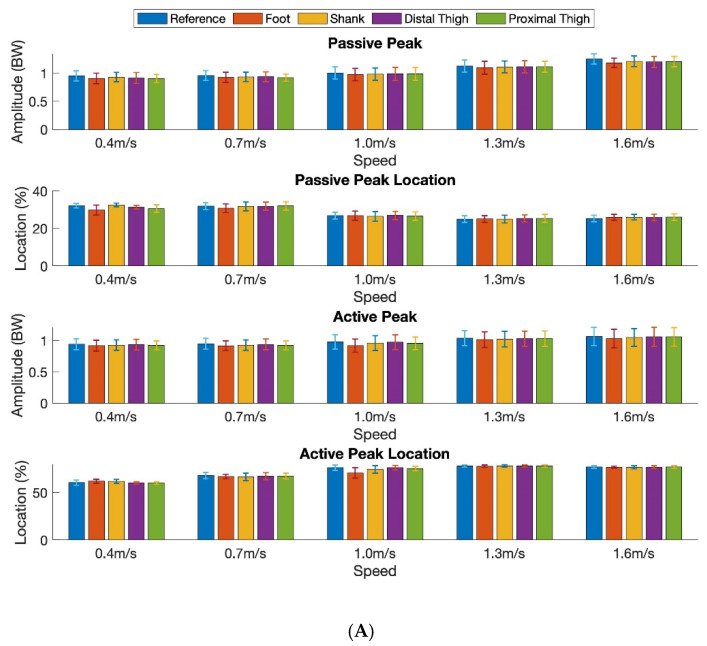
Mean peak amplitude of the reference and estimated vGRF (passive and active force peaks) and locations averaged across all nine participants for each sensor location and each speed in intra-participant testing (**A**) and inter-participant testing (**B**). The amplitude of the vGRF is normalized to body weight (BW).

**Table 1 sensors-20-04345-t001:** Mean intra-participant testing accuracies averaged from all five walking speeds across all participants, for all four wearing locations, respectively.

	IMU Position			
	Foot	Shank	Distal Thigh	Proximal Thigh
R	0.99 (0.00)	1.00 (0.00)	0.99 (0.00)	0.99 (0.03)
RMSE (BW)	0.03 (0.01)	0.02 (0.01)	0.02 (0.01)	0.03 (0.03)
NRMSE	2.33% (0.53%)	1.70% (0.39%)	1.75% (0.63%)	2.31% (1.83%)
PP error	0.04 (0.02) BW ^-a^*	0.02 (0.01) BW ^-b^*	0.03 (0.01) BW	0.03 (0.02) BW
PP delay	0.53% (1.07%)	−0.14% (0.40%)	−0.05% (0.48%)	0.15% (0.79%)
AP error	0.04 (0.01) BW ^-a^*	0.02 (0.00) BW ^-b^*	0.01 (0.00) BW	0.01 (0.01) BW
AP delay	1.04% (2.36%)	0.45% (1.08%)	0.15% (0.23%)	0.14% (0.44%)

Note: peak error and peak delay are derived by subtracting the estimated peak amplitude and location (relative to the percentage of stance duration) from the corresponding reference values. R, correlation coefficient; RMSE, root mean squared error; NRMSE, normalized root mean square error; PP, passive peak; AP, active peak. ^-a^*^, -b^* items labeled ^-a^* are significantly different from items labeled ^-b^* in the same row, but there is no significant difference within ^-a^*^, -b^* items, respectively.

**Table 2 sensors-20-04345-t002:** Mean inter-participant testing accuracies averaged from all five walking speeds across all participants, for all four wearing locations, respectively.

	IMU Position			
	Foot	Shank	Distal Thigh	Proximal Thigh
R	0.94 (0.02)	0.97 (0.02) ^-a^*	0.92 (0.02) ^-b^*	0.95 (0.05)
RMSE (BW)	0.12 (0.04)	0.10 (0.03) ^-a^*	0.14 (0.04) ^-b^*	0.12 (0.03)
NRMSE	8.42% (3.32%)	7.15% (2.03%) ^-a^*	9.95% (2.55%) ^-b^*	8.71% (2.19%)
PP error	0.25 (0.13) BW ^-a^*	0.14 (0.04) BW ^-b^*	0.24 (0.05) BW	0.16 (0.05) BW
PP delay	2.72% (5.52%)	0.57% (2.54%)	4.20% (5.82%)	2.16% (3.67%)
AP error	0.16 (0.06) BW ^-a^*	0.08 (0.02) BW ^-b^*	0.09 (0.04) BW	0.09 (0.06) BW
AP delay	4.24% (4.22%)	1.66% (2.89%)	2.03% (2.59%)	2.28% (2.26%)

Note: peak error and peak delay are derived by subtracting the estimated peak amplitude and location (relative to the percentage of stance duration) from the corresponding reference values. R, correlation coefficient; RMSE, root mean squared error; NRMSE, normalized root mean square error; PP, passive peak; AP, active peak. ^-a^*^, -b^* items labeled ^-a^* are significantly different from items labeled ^-b^* in the same row, but there is no significant difference within ^-a^*^, -b^* items, respectively.

**Table 3 sensors-20-04345-t003:** Mean intra-participant testing accuracies obtained from each of the five walking speeds across all participants with IMUs of all four locations.

Speeds (m/s)					
	0.4	0.7	1.0	1.3	1.6
R	0.99 (0.01)	0.98 (0.04)	0.99 (0.00)	1.00 (0.00)	0.99 (0.00)
RMSE (BW)	0.04 (0.02)	0.03 (0.03)	0.03 (0.01)	0.02 (0.01)	0.03 (0.01)
NRMSE	2.60% (1.04%)	2.17% (1.53%)	1.81% (0.64%)	1.77% (0.49%)	2.09% (0.71%)
PP error	0.04 (0.02) BW ^-a^*	0.03 (0.01) BW ^-a^*	0.02(0.01)BW ^-a^*	0.02 (0.01) BW ^-a^*	0.05 (0.03) BW ^-b^*
PP delay	1.05% (0.95%) ^-a^*	0.44% (0.89%)	0.11% (1.3%) ^-b^*	−0.23% (0.34%) ^-b^*	−0.71% (0.57%) ^-b^*
AP error	0.02 (0.01) BW ^-a^*	0.02 (0.01) BW ^-b^*	0.03(0.02)BW ^-b^*	0.01 (0.01) BW ^-b^*	0.02 (0.01) BW ^-b^*
AP delay	−0.57% (2.55%) ^-a^*	0.7% (2.03%)	2.05%(2.68%) ^-b^*	0% (0.25%) ^-a^*	0.13% (0.24%) ^-a^*

Note: peak error and peak delay are derived by subtracting the estimated peak amplitude and location (relative to the percentage of stance duration) from the corresponding reference values. R, correlation coefficient; RMSE, root mean squared error; NRMSE, normalized root mean square error; PP, passive peak; AP, active peak. ^-a^*^, -b^* items labeled ^-a^* are significantly different from items labeled ^-b^* in the same row, but there is no significant difference within ^-a^*^, -b^* items, respectively.

**Table 4 sensors-20-04345-t004:** Mean inter-participant testing accuracies obtained from each of the five walking speeds across all participants with IMUs of all locations.

Speeds (m/s)					
	0.4	0.7	1.0	1.3	1.6
R	0.94 (0.01)	0.94 (0.03)	0.95 (0.02)	0.96 (0.02)	0.95 (0.02)
RMSE(BW)	0.12 (0.03)	0.12 (0.04)	0.11 (0.03)	0.11 (0.03)	0.13 (0.02)
NRMSE	9.40% (2.81%)	8.87% (3.45%)	8.58% (2.62%)	8.24% (2.39%)	9.38% (2.52%)
PP error	0.19 (0.10) BW	0.14 (0.12) BW ^-a^*	0.16 (0.14) BW^-a^*	0.20 (0.14) BW	0.30 (0.06) BW ^-b^*
PP delay	9.24% (3%)^a^*	5.06% (3.72%) ^a^*	0.87% (2.98%)^-b^*	−0.78% (1.86%)^-b^*	−2.26% (1.17%) ^-b^*
AP error	0.06 (0.11) BW	0.07 (0.13) BW	0.11 (0.13) BW	0.12 (0.14) BW	0.17 (0.11) BW
AP delay	−2.24% (3.08%) ^-a^*	3.69% (3.85%) ^-b^*	6.46% (3.03%)^-b^*	3.65% (2.65%)^-b^*	1.39% (2.32%) ^-b^*

Note: peak error and peak delay are derived by subtracting the estimated peak amplitude and location (relative to the percentage of stance duration) from the corresponding reference values. R, correlation coefficient; RMSE, root mean squared error; NRMSE, normalized root mean square error; PP, passive peak; AP, active peak. ^-a^*^, -b^* items labeled ^-a^* are significantly different from items labeled ^-b^* in the same row, but there is no significant difference within ^-a^*^, -b^* items, respectively.

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
