# Peer review of "Estimating Vertical Ground Reaction Force during Walking Using a Single Inertial Sensor"

_sensors, 2020, doi:10.3390/s20154345_

Round 1

Reviewer 1 Report

The article describes the use of random forest for the prediction of vertical ground reaction forces in walking. The paper is concise and matches with the scope of the MDPI Sensors journal. The work overall has rationality and completeness of research problem and sufficient explanation of findings. The methods and discussion are appropriate but require further details and justification. References however are unfitting and require significant changes (see comments to authors). Language quality is at the academic level but the document’s organization may need improvement.

I have the following criticism for the better development and improvement of the paper. Also, the suggested articles may be referred, but the authors are not forced to do so.

  • Page 2, line 53. None of the cited articles [4-9] describe load cell or pressure insole sensors for the prediction of GRFs.
  • Page 2, line 59: The authors clearly have made a mistake with their citations, they refer to a publication by Karatsidis et al. [17] but the 17th reference on the list is by Lipfert et al. Also, its clearly not a numbering mistake since there is not a paper by Karatsidis in the reference list.
  • Page 2, line 62: as above, for the reference of Shahabpoor et al. [18]. I found multiple similar mistakes throughout the text, authors should revise their entire reference list.
  • Page 2, line 78: the authors state that “machine learning methods have made it possible to approach gold standard (force plate) estimation”. To my knowledge, none of the existing reports managed to present predictions with RMS errors smaller than 0.1 BMI. The authors claimed that their “approach has achieved among the best performances” but they report errors of approximately 0.12 BMI. Thus, the statement that ML estimations are comparable to force platform recordings seems implausible.
  • Page 2, line 81: “a single inertial sensor was able to estimate the magnitude and timing of the vGRF peaks during walking”, I found this statement confusing since the authors used one IMU to predict the GRF of a single leg. It may be more precise to state that a single IMU is needed per lower limb.
  • Page 2, lines 81-83: the use of a random forest model for GRF predictions is indeed novel, but there are multiple reports in the literature describing ML models for the prediction of GRF in walking with a single IMU or for the optimisation of sensor placement. The authors could potentially refer to such articles and highlight the novelty of their approach. For example:

Tian Tan, David P. Chiasson, Hai Hu, Peter B. Shull, Influence of IMU position and orientation placement errors on ground reaction force estimation, Journal of Biomechanics, Volume 97, 2019,

Lim, H.; Kim, B.; Park, S. Prediction of Lower Limb Kinetics and Kinematics during Walking by a Single IMU on the Lower Back Using Machine Learning. Sensors 2020, 20, 130.

  • Page 4, line 117: “linear fit, and parabolic fit”. I presume you refer to the coefficients of liner and parabolic fits.
  • Page 5, line 171-175: Mixture of methods with results. This section belongs in methods.
  • Page 6 and 9, Figures 3 and 5: it may be better to plot predictions and reference waveforms on the same graph for better visual comparisons.
  • Page 9, lines 241, section 3.2. Inter-participant testing. The entire section 3.2. is very repetitive and the wordings is identical to section 3.1. It may be better to merge sections 3.1. and 3.2 together or rephrase entirely.
  • Page 12, lines 285-287: “Our approach has achieved among the best performances when compared to other methods in the literature”. There might be some room here to compare your results with other similar approaches and their findings.

Author Response

The authors thank the reviewer for the valuable comments and suggestions. We have carefully read each of these questions and have addressed them with the answers and corresponding changes in the manuscript (revision tracked). All line numbers referred below are with word revision tracking on.

Question 1: The article describes the use of random forest for the prediction of vertical ground reaction forces in walking. The paper is concise and matches with the scope of the MDPI Sensors journal. The work overall has rationality and completeness of research problem and sufficient explanation of findings. The methods and discussion are appropriate but require further details and justification. References however are unfitting and require significant changes (see comments to authors). Language quality is at the academic level but the document’s organization may need improvement.

Answer 1: Thank you for your valuable comments. We have fixed the reference problems and improved document organization according to your suggestions. See details in the point-to-point answers following.

Question 2:

  • Page 2, line 53. None of the cited articles [4-9] describe load cell or pressure insole sensors for the prediction of GRFs.

Answer 2:

Thanks for pointing out the mistake. The references have been updated see line 113.

Question 3:

  • Page 2, line 59: The authors clearly have made a mistake with their citations, they refer to a publication by Karatsidis et al. [17] but the 17th reference on the list is by Lipfert et al. Also, its clearly not a numbering mistake since there is not a paper by Karatsidis in the reference list.

Answer 3:

Thanks for pointing out the mistake. The references have been updated [12] see line 118.

Question 4:

  • Page 2, line 62: as above, for the reference of Shahabpoor et al. [18]. I found multiple similar mistakes throughout the text, authors should revise their entire reference list.

Answer 4:

Thanks for pointing out the mistake. The references have been updated [13] see line 122.

We have checked and revised the whole the reference list (there was an error caused by our reference manager system).

Question 5:

  • Page 2, line 78: the authors state that “machine learning methods have made it possible to approach gold standard (force plate) estimation”. To my knowledge, none of the existing reports managed to present predictions with RMS errors smaller than 0.1 BMI. The authors claimed that their “approach has achieved among the best performances” but they report errors of approximately 0.12 BMI. Thus, the statement that ML estimations are comparable to force platform recordings seems implausible.

Answer 5:

Thank you for drawing our attention to this point. We have now specifically cited the two similar studies (instead of citing the Ancillao systematic review) that have used a ML approach to estimate GRF in walking with IMUs (line 139 and line 660). Guo et al used an IMU to estimate GRF in walking and reported an average prediction error of 3.8% in intra-participant analysis. Leborace et al reported errors of 4.7-4.8% in walking. Errors of 4-5% correspond to 0.04-0.05 BWs. These results are in comparison to our intra-participant error of 1.70% and 0.02 BW for the IMU located on the shank (the error in our inter-participant analysis was higher at 7.15%).

Question 6:

  • Page 2, line 81: “a single inertial sensor was able to estimate the magnitude and timing of the vGRF peaks during walking”, I found this statement confusing since the authors used one IMU to predict the GRF of a single leg. It may be more precise to state that a single IMU is needed per lower limb.

Answer 6:

The sentence in the text (line 142) has been revised to “a single inertial sensor per lower limb was able to estimate the magnitude and timing of the vGRF peaks during walking”

Question 7:

  • Page 2, lines 81-83: the use of a random forest model for GRF predictions is indeed novel, but there are multiple reports in the literature describing ML models for the prediction of GRF in walking with a single IMU or for the optimisation of sensor placement. The authors could potentially refer to such articles and highlight the novelty of their approach. For example:
    • Tian Tan, David P. Chiasson, Hai Hu, Peter B. Shull, Influence of IMU position and orientation placement errors on ground reaction force estimation, Journal of Biomechanics, Volume 97, 2019,
    • Lim, H.; Kim, B.; Park, S. Prediction of Lower Limb Kinetics and Kinematics during Walking by a Single IMU on the Lower Back Using Machine Learning. Sensors 2020, 20, 130.

Answer 7:

Thanks for pointing out the missing reference, we have added the following text at line 141, “, as IMUs orientation and placement would affect the estimation of vGRF [24,25]”

Question 8:

  • Page 4, line 117: “linear fit, and parabolic fit”. I presume you refer to the coefficients of liner and parabolic fits.

Answer 8:

Yes, the original line 117 has been changed to “the coefficients of linear fit and parabolic fit” (line 206)

Question 9:

  • Page 5, line 171-175: Mixture of methods with results. This section belongs in methods.

Answer 9:

The first sentence originally at line 171 was removed, and “(age 27 ± 8; height 176 ± 7 cm; weight 72 ± 9 kg)” has been added to line 175 in the methods part.

Question 10:

  • Page 6 and 9, Figures 3 and 5: it may be better to plot predictions and reference waveforms on the same graph for better visual comparisons.

Answer 10:

Thank you for your suggestion. We have tried the same idea as yours before but found out it was not as good as the present presentation with the concern that too many curves are in a single plot.

Question 11:

  • ?Page 9, lines 241, section 3.2. Inter-participant testing. The entire section 3.2. is very repetitive and the wordings is identical to section 3.1. It may be better to merge sections 3.1. and 3.2 together or rephrase entirely.

Answer 11:

The section 3.1 and 3.2 have been merged.

Question 12:

  • ?Page 12, lines 285-287: “Our approach has achieved among the best performances when compared to other methods in the literature”. There might be some room here to compare your results with other similar approaches and their findings.

Answer 12:

Thanks for the comment. We have added some discussion (659-661) comparing the results from other similar studies and how they compare to our findings.

Reviewer 2 Report

This study attempted to determine if a single IMU could be used to estimate the magnitude and timing of the vertical ground reaction force peaks during walking and also to find the best location of four possible locations on the lower extremity for the IMU.  Overall, this is a very well-written report of a clearly-defined research protocol.  The study design was appropriate and the description of the methods and analyses clearly outlined.  Admittedly, some of the data processing techniques were novel to this reviewer but, given the expertise of the research team, one can assume that these were the most appropriate choices for the task.

There really aren't any issues with the manuscript that would prevent its publication.  While the classification of the two peaks in the vertical GRF as being passive and active may be common, I think the way the authors described these in lines 42-44 could be revised slightly.  The impression one gets from what is written is that the passive peak is simply an impact artefact which ignores the eccentric muscle activity that goes into preventing the collapse of the supporting limb and also fails to acknowledge that while the early part of vGRF is happening for the ipsilateral limb, the active portion is occurring on the contralateral limb.  I suppose what I mean is that the word "passive" may be a little misleading even though it may be common in the literature.  The GRF is the result of all the mass-acceleration products of the segments of the body, so the description of 'passive' is, in my opinion, problematic.  Perhaps, the initial peak could be better described as the "weight acceptance peak"?

I would also be curious to know if the technique they are using might work to estimate the two shear GRFs as well.  The sensors have 9 degrees of freedom so the other two axes chould be available; however, I am not familiar enough with the post-processing techniques to know if these other axes are available (or are these to appear in future submissions, perhaps?). 

Author Response

The authors thank the reviewer for the valuable comments and suggestions. We have carefully read each of these questions and have addressed them with the answers and corresponding changes in the manuscript (revision tracked). All line numbers referred below are with word revision tracking on.

Question 1:

While the classification of the two peaks in the vertical GRF as being passive and active may be common, I think the way the authors described these in lines 42-44 could be revised slightly.  The impression one gets from what is written is that the passive peak is simply an impact artefact which ignores the eccentric muscle activity that goes into preventing the collapse of the supporting limb and also fails to acknowledge that while the early part of vGRF is happening for the ipsilateral limb, the active portion is occurring on the contralateral limb.  I suppose what I mean is that the word "passive" may be a little misleading even though it may be common in the literature.  The GRF is the result of all the mass-acceleration products of the segments of the body, so the description of 'passive' is, in my opinion, problematic.  Perhaps, the initial peak could be better described as the "weight acceptance peak"?

Answer 1:

Thank you for your comments. We share your concerns around the terminology used to describe the two vGRF peaks and decided on referring to them as the passive and active peaks based on our review of the literature. While the initial peak could be called the “weight acceptance peak,” the preferred term in the literature is “passive peak” as the majority of the vGRF generated is due to the passive impact with the ground after initial contact while the second peak is mainly due to the active force generated to push back into the ground towards toe-off.

Question 2:

I would also be curious to know if the technique they are using might work to estimate the two shear GRFs as well.  The sensors have 9 degrees of freedom so the other two axes chould be available; however, I am not familiar enough with the post-processing techniques to know if these other axes are available (or are these to appear in future submissions, perhaps?).

Answer 2:

Thank you for your comments. The anteroposterior and mediolateral components could also be estimated with this method and it would be an interesting further investigation. We chose to limit our analysis to the vertical component in this study, instead focusing on the accuracy across speeds and IMU locations.